# Performance Analysis of Deep Learning Algorithms in Diagnosis of Malaria Disease

**DOI:** 10.3390/diagnostics13030534

**Published:** 2023-02-01

**Authors:** K. Hemachandran, Areej Alasiry, Mehrez Marzougui, Shahid Mohammad Ganie, Anil Audumbar Pise, M. Turki-Hadj Alouane, Channabasava Chola

**Affiliations:** 1Department of Analytics, School of Business, Woxsen University, Hyderabad 502345, Telangana, India; 2College of Computer Science, King Khalid University, Abha 62529, Saudi Arabia; 3Siatik Premier Google Cloud Platform Partner, Johannesburg 2000, South Africa; 4School of Computer Science and Applied Mathematics, University of the Witwatersrand, Johannesburg 2000, South Africa; 5School Saveetha School of Engineering, Chennai 600124, Tamil Nadu, India; 6Department of Studies in Computer Science, University of Mysore, Manasagangothri, Mysore 570006, Karnataka, India

**Keywords:** deep learning techniques, convolution neural networks, ResNet-50, mobilenet, disease diagnosis, malaria, neural networks and RBC

## Abstract

Malaria is predominant in many subtropical nations with little health-monitoring infrastructure. To forecast malaria and condense the disease’s impact on the population, time series prediction models are necessary. The conventional technique of detecting malaria disease is for certified technicians to examine blood smears visually for parasite-infected RBC (red blood cells) underneath a microscope. This procedure is ineffective, and the diagnosis depends on the individual performing the test and his/her experience. Automatic image identification systems based on machine learning have previously been used to diagnose malaria blood smears. However, so far, the practical performance has been insufficient. In this paper, we have made a performance analysis of deep learning algorithms in the diagnosis of malaria disease. We have used Neural Network models like CNN, MobileNetV2, and ResNet50 to perform this analysis. The dataset was extracted from the National Institutes of Health (NIH) website and consisted of 27,558 photos, including 13,780 parasitized cell images and 13,778 uninfected cell images. In conclusion, the MobileNetV2 model outperformed by achieving an accuracy rate of 97.06% for better disease detection. Also, other metrics like training and testing loss, precision, recall, fi-score, and ROC curve were calculated to validate the considered models.

## 1. Introduction

Malaria due to Plasmodium falciparum is life-threatening and is still a matter of concern in the world. Every year hundreds of millions of people are found to be affected by this disease as per the report by WHO (World Health Organization). In the year 2022 [1], approximately 247 million malaria cases were reported. This disease is spread by mosquitoes (female anopheles), biting between dusk and dawn. They are also known as “night-biting” mosquitoes. For the detection of malaria disease, there are a few techniques practiced named rapid diagnostic test (RDT) [2], clinical diagnosis, polymerase chain reaction (PCR) [3], and microscopic diagnosis. The efficiency of the conventional methods of diagnosis like clinical and PCR depend on the ability of the available human.

The other two methods (RDT and microscopic diagnosis) are practiced because they contribute the most to controlling malaria nowadays as they are effective diagnosis methods for malaria [4]. The RDT method used for diagnosis is remarkably effective due to no demand for experts. The microscopic systems require expert microscopists, considered an effective diagnosis method as there are no shortcomings as in RDT. The involvement of microscopic blood smear images in malaria detection known as automatic microscopic can be an effective tool for diagnosis as it segments the cells and classifies the infected cells [5,6]. A modified Convolution network was designed for the visual recognition challenge [7,8]. Rectified-Correlations on a Sphere (RECOS) are proposed with CNN for a multilayer system with AlexNet for the MNIST dataset [9]. The European Bioinformatics Institute concentrated on the data by increasing the bandwidth and computational infrastructure, because more data is required to increase the prediction accuracy [10].

In 2018, M.Z. Alom surveyed deep learning approaches, in that survey, they discussed advanced techniques for training models and other generative models [11]. For the autonomous driving application, the GoogleNet model performance is better compared with CNN [12]. Some generalized models such as Bayesian, linear regression with Relu, and Support Vector Machine are frequently used as common prediction and classification models [13,14,15,16,17]. In 2019, malaria disease was predicted using the CNN model with an accuracy of 90% [18,19]. In other studies, mobile applications were used for predicting malaria disease using CNN [20,21]. Yuhang Dong designed a model that automatically learns the features from the input for diagnosing malaria disease using CNN [22,23]. Semantic Segmentation is a process of extracting minute information from any Image dataset for classification [24,25,26,27]. CNN’s are effective methods for identifying picture classes [28]. Deep learning has been used in very few studies to differentiate types with cryptic morphological variation. Convolutional neural networks (CNNs) are deep learning algorithms that are good at extracting features from an organized array of data like photos [29]. These features include patterns like lines, gradients, contours, and geometries. Neural networks, like CNN, have the benefit of being able to identify features from raw image datasets [30]. In contrast with typical machine learning algorithms, which extract certain features and feed them into the training algorithm, CNN extracts the necessary features by itself. Mobile Net V2 is a compact CNN model designed for mobile machine learning deployment. The MobileNet concept aims to maximize computational power and memory use.

However, there is a lack of study around classification with techniques like image processing, machine learning, and computer vision, these tools provide accurate and efficient computation in detecting malaria disease. This contribution towards malaria parasite detection will play a significant role in building an automated system with high accuracy for the future. The main contributions of our work are as follows:A new framework for malaria disease diagnosis is proposed based on deep learning techniques like CNN, MobileNetV2, and ResNet50.Employment of hyperparameters and fine-tuning to improve the overall classification performance to identify the infected and uninfected cell image of malaria disease.Validation of different performance metrics is done to identify the reliability and feasibility of the proposed framework.

## 2. Related Works

The study of malaria patients in Bangladesh for improving malaria detection by rapid diagnostic test (RDT), using commercially available RDTs is not satisfactory when compared to the real-time PCR assay [2]. However, malaria leads to deaths and is a major concern for underdeveloped countries, primarily diagnosed with the help of microscopy. Diagnosis of malaria can be done through a computer that separates the image sample of the infected and uninfected class by the appearance of the image [7].

An automated CNN-based model is remarkably effective for diagnosis as it provides accurate results [8] with the help of the CNN model. For example, Dong et al. have compared the outcomes of three widely used Convolutional Neural Networks (CNN) [8], a few to be named, LeNet-5 [9], AlexNet [11], and GoogLeNet [12]. CNN uses a few techniques for diagnosis which include data augmentation, autoencoders, and feature extraction. These are classified by K-Nearest Neighbors (KNN) or Support Vector Machine (SVM). Quinn et al. [14] and Rosado et al. [31] both addressed the issue of model computational efficiency. However, in the quest for computing productivity, both investigations discovered a significant deterioration in the accuracy of the model. Rosado et al. have studied the recognition of malaria parasites as well as white blood cells (WBC) using cell phones. By the usage of a Support Vector Machine (SVM), they have reported sensitivity and specificity of 80.5% and 93.8% for trophozoite identification and 98.2% and 72.1% for white blood cells (WBC), respectively. The model’s accuracy and performance were optimized and improved by training procedures general, distillation, and autoencoder training. With the help of the deep learning model, the results achieved are realistic by metric analysis like an accuracy of 99.23% experienced for visual recognition [8,32]. With their 16-layered CNN model, Liang et al. achieved 97.37% accuracy in detecting the disease, claiming that his model is better than transfer learning approaches [9].

Apart from the models mentioned, in recent years Bayesian decision networks (BDNs) model has become a more predominant method [13]. They have shown much better prediction when compared to other traditional methods. Predictive analysis can also be determined using generalized linear models (GLM) which consider many factors like demographic, and environmental covariates [14]. But the relations amongst variables and correlation structures in GLM differ across a study area or for a period. BDNs are most prominently used because of better predictive performance and a more flexible displaying system than GLMs. Profound Learning models, or to be more explicit, Convolutional Neural Networks (CNNs) have been demonstrated to be truly compelling in a wide variety of works and hence we here prefer to continue the work in the CNN model [18]. Alex Krizhevsky and his team trained a large Deep-CNN to classify images in the ImageNet LSVRC-2010 and 2012 [33,34]. In their model, they used 650,000 neurons with GPU and other parameters such as kernel and spectral algorithm. In the 2012 contest, they achieved the best result with an error rate of 15.3% [35,36].

In 2019, Rajaram S and his team built an ensemble model for classification. This ensemble method is a combination of both VGG-19 and SqueezeNet. Through this methodology, the variance, and overfitting of the model are reduced [37]. Julien Reboud explained the DNA-based sensors for diagnosing malaria disease without visiting the laboratory [38]. In the same year, Seble Girma conducted a study on diagnosing malaria disease, and identified that uRDT Alere Malaria has a high sensitivity to RDT and microscopy in detecting asymptomatic malaria [39].

In 2020, Femi Emmanuel Ayo built a decision support system for multi-target disease diagnosis. But he faced difficulties in finding the difference between malaria and typhoid. So, finally, he used t-test statistics for calculating the mean values of diagnosis. Through that, his proposed diagnosis system got a higher accuracy [40]. A mobile system for healthcare was built by utilizing CNN with a stochastic gradient descent optimizer for detecting malaria. The accuracy is fine. But if new data is given to this model, the accuracy is drastically reduced because of bias or overfitting the model [41]. Brian Gitta and Rosauro Varo say that microscopic examination of Giemsa-stained blood films is very efficient in plasmodium detection. The authors have done a study on plasmodium detection, and he also discussed the merits and demerits of other diagnostic methods in the market [42,43].

Rupam Srivastava 2022 proposed a system for diagnosing malaria disease. In this system, the fiber sensor is far better than the traditional systems [44]. Jianhai Yin conducted a literature study on the detection of malaria in laboratories in China. In this paper, the author provides an overview of China’s laboratory networks, quality control, and diagnostics throughout the elimination phase [45]. In this study the fundamentals, effectiveness, shortcomings, successes, and application of a variety of malaria diagnostic methods as well as their potential contribution to lowering the prevalence of malaria in Sub-Saharan Africa [46]. Since the malaria parasite may change the host’s microRNA (miRNA) expression, miRNAs can aid in the detection of malaria infection. The molecular diagnostic tool that can identify diseases may identify these mutations. We provide an overview of what is currently known about miRNA during malaria infection. Initial studies have revealed the potential for using miRNAs as biomarkers to diagnose and treat several disorders, including malaria [47]. In areas with high levels of transmission, it might be challenging to appropriately identify severe Plasmodium falciparum malaria in children. They calculated that detecting 150,000 platelets/L and a plasma P. falciparum histidine-rich protein 2 (PfHRP2) concentration had a sensitivity of 74% and a specificity of 93% in identifying severe malaria. This was done by fitting Bayesian latent class models using a combination of platelet counts and P. falciparum histidine-rich protein 2 (PfHRP2) plasma concentrations. These results showed that some kids participating in clinical research on malaria in areas with high transmission also suffer severe febrile illnesses brought on by other diseases [48].

In 2022, Mariki gathered Data from Morogoro and Kilimanjaro in Tanzania to diagnose malaria. Mariki has built a random forest machine learning model and trained the model with the data. Based on the training, the model has an accuracy of 95% in Kilimanjaro, 87% in Morogoro, and 82% throughout the entire dataset, Random Forest was the most accurate classifier. The regional-specific malaria-predicting model was created based on demographic data and clinical symptoms to illustrate useful machine-learning classifiers [49]. Goncalo Marques utilized convolutional neural networks to identify malaria from microscopic pictures of peripheral blood cells (CNN). The EfficientNetB0 architecture is the foundation of the suggested paradigm. According to his research, Efficient Net is a trustworthy architecture for malaria-automated medical diagnosis [50]. During the training the performance was fine. But when validation data is passed into the model the performance of the model decreased.

## 3. Materials and Methods

In this section, we discuss the technical aspects of our work, including the dataset definition, the proposed architecture, and its components. Different deep learning techniques like convolutional neural networks (CNN), MobileNetV2, and ResNet50 were employed for better classification of malaria disease.

### 3.1. Dataset Description

This dataset was taken from the National Institutes of Health (NIH) website (https://www.nih.gov/, accessed on 11 August 2022) and contains 27,558 photos, including 13,780 parasitized cell images and 13,778 uninfected cell images. The National Institutes of Health (NIH) is made up of 27 institutes and centers, each with its research agenda, frequently focusing on specific diseases or bodily systems hence we have explored the data set for malaria disease from this online repository. The infected and uninfected cell images are shown in Figure 1 and Figure 2.

### 3.2. The Convolution Neural Networks (CNN) Model

A convolutional neural network (CNN) is a subclass of deep learning that uses a variant of the multilayer perceptron to achieve the required minimum preprocessing. CNN’s are feed-forward artificial neural networks. CNN simplifies the process of establishing effective feature extraction and classification of malaria, which requires domain expertise [7]. It contains many processing layers that use image analysis filters shown in Figure 3.

In Figure 4, rectified linear activation function (ReLU) gives the output based on the input, if the input is positive, it directly provides the output, otherwise, it will output zero. Convolution Layer-Work of a convolution layer is to transform the input image and extract the features from it. The image is convolved with a kernel or filter while the transformation process and then provides the desired output.

Input Value—a

Bias—b

Weight—w

The input value consists of the image’s raw input data and convolution layer is the primary component of CNN architecture. The convolution is a very fundamental process of applying a filter to an input (a in our case) to produce an activation. When the same filter is applied to an input multiple times, a feature map is created, displaying the positions and strength of a recognized feature in an input, such as an image (like malaria images). It is further subjected to the activation function RelU (Rectifies Linear Unit). It is a piece-wise linear function that will result in output to the user’s input only if it is positive, and zero otherwise [51]. Since a model that utilizes it is quicker to train and produces higher performance, this has become the algorithm’s preferred activation function for several machine learning algorithms this process is repeated until we get the desired output. The equations used to provide the output are given as:(1)$$a_{l+1}=f\;(w_{l}+a_{l})+b,a_{l}$$
(2)$$a_{l+2}=f\;(w_{l+1}+a_{l+1})+b,a_{l+1}a_{l}$$

### 3.3. The MobileNetV2 Model

MobileNet is a compact and lightweight deep neural network with fewer parameters and higher classification performance than other deep neural networks. Dense blocks, as proposed and projected in Dense Nets, are integrated into MobileNet to minimize the number of network parameters, and increase the accuracy of classification. Convolution layers of the very same dimensions as the input feature maps in MobileNet networks are often used as compact blocks in Dense-MobileNet models, where compact interconnections are accomplished within these blocks shown in Figure 5. The network cuts down the constraints and computation costs even further by setting a little growth rate. Dense1-MobileNet and Dense2-MobileNet are two types of Dense-MobileNet models [52]. Experimentations revealed that Dense2-MobileNet can outperform MobileNet in terms of recognition accuracy while using fewer parameters and computing resources.

### 3.4. The ResNet50 Model

RESNET (residual neural network) was developed at Microsoft Research Institute and proposed by the Kaiming team [53]. It was specifically used to solve the degradation problems of different neural network architectures, that is when the hidden layer increases the training error also increases. To solve this problem, this team has proposed the residual structure. The residual function is programmed in this architecture. This function is used to calculate the error between the actual value and the estimated/predicted value. RESNET (Residual Network) has several kinds of architectures but in this study ResNet50 as shown in Figure 6 is used to develop a framework for detecting disease. The residual component is composed of two convolution layers and an identity mapping. The convolution kernel size is 3 × 3. Therefore, the input and output dimensions of the residual component are the same and can be added directly.

## 4. Experimental Results

Malaria is a serious disease that continues to cause concern around the world. Three deep learning models CNN, MobileNetV2, and ResNet50 were compared in this paper. This study uses an experimental design technique to examine the efficiency of new deep-learning methods for malaria identification using blood smear pictures. We used models to infer post-training the deep learning architectures with the training data and then finally validate the trained models for better detection of disease.

Table 1 presents the model summary of the CNN model. The layer (type), output shape, and the parameters of the training phase of convolutional neural networks have been presented.

The classification performances of the algorithms were evaluated using a confusion matrix. The confusion matrices of CNN MobileNetV2 and ResNet50 models are shown in Figure 7, Figure 8 and Figure 9. In this experiment, MobileNetV2 outperformed by attaining the highest accuracy rate of 97.06%.

The considered models were trained and tested by changing the epoch and batch size along with fine-tuning various hyperparameters. The algorithm’s epoch number is a crucial hyperparameter. It describes the number of epochs or full passes through the process’s training or learning phase for the entire training dataset.

Figure 10 presents the accuracy and loss for training and validation of the CNN model. The training accuracy is 94.92% and the testing accuracy is 95.90%. Where the green line demonstrates the accuracy and loss in the training phase while the red line demonstrates the accuracy and loss in the validation phase [54]. It can be inferred from the graph that the training and validation loss of the CNN model is 0.16% and 0.15%, respectively. In addition, Figure 11 depicts the accuracy and loss for training and validation of the MobileNetV2 model. It achieved the highest accuracy 99.90% for training and 97.06% for validation. In addition, it can be inferred from the graph that the training and validation loss of the MobileNetV2 model is 9.71% and 0.17% respectively. In Figure 12, the accuracy and loss for training and validation of the ResNet50 model are shown. It attained the training and validation accuracy of 99.60% and 96.70% respectively. Also, it can be inferred from the graph that the training and validation loss of the ResNet50 model is 0.001% and 0.166% respectively.

The ROC curve was used to show the prediction ability of the CNN, MobileNetV2, and ResNet50 algorithms at different thresholds, as shown in Figure 13, Figure 14 and Figure 15. It represents the false-positive rate (FPR) vs. the true-positive rate (TPR) along the *x*-axis and *y*-axis, respectively. A larger ROC area suggests the model’s distinguishing ability between 0′s and 1′s, leading to a better prediction. An AUC value closer to Also, an AUC value closer to 1 denotes a good separability measure, while an AUC value close to 0 denotes the worst measure of disassociation. In the case of an AUC of 0.5, the model becomes ineffective in separating the classes [55]. It can be observed that the ResNet50 model performs best with a ROC curve value of 96.73 followed by MobileNetV2, and CNN with values of 96.66, and 95.91 respectively.

Other statistical measurements like precision, recall, and f1-score were calculated to validate the developed models for malaria detection. In terms of precision and recall, ResNet50 achieved the highest rate of 0.97%. While, in terms of f1-score, MobileNetV2 and ResNet50 attained a rate of 0.97%.

In addition to accuracy, we calculated the precision, recall, and f1-score of the employed models as shown in the Figure 16, Figure 17 and Figure 18 respectively. In addition, the micro average and the weighted average were measured for both classes (uninfected and infected).

## 5. Discussion

Malaria is a serious disease that continues to cause concern around the world. Three deep learning models CNN MobileNetV2, and ResNet50 were compared in this paper. This study uses an experimental design technique to examine the efficiency of new deep-learning methods for malaria identification using blood smear pictures. We used models to infer post-training the deep learning architectures with the training data.

We used 25 epochs in all the considered models, in which an epoch is a whole transit of the learning data through the machine learning (ML) algorithm. The epoch number of the process is a critical hyperparameter. The entire training dataset specifies the number of epochs or full pass through the algorithm’s training or learning phase. A few of the observations that are made from the results are that the above figures are related to loss and accuracy in terms of training and testing of the CNN model, MobileNetV2, and ResNet50, respectively. It can be seen after comparing all the graphs that the training accuracy of the MobileNetV2 is higher than CNN and ResNet50 models and that the validation accuracy of the ResNet is higher than that of other considered models. It can be observed that the training error of the ResNet50 is less compared to others. It can also be observed from above that in CNN model is having a significant drop in the training and validation error. Based on the accuracy and error we can conclude that ResNet50 is best as the accuracy was high and the error was less as compared to the CNN and MobileNetV2 models.

## 6. Conclusions

In this paper, we compare three deep learning techniques like CNN, MobileNetV2, and ResNet50 in terms of detecting malaria disease. The developed models were compared, and conclusions were reached based on which was superior. It can be concluded that environmental elements are crucial in facilitating. Malaria’s presence and transmission. Among all of the models, ResNet50 outperformed and provided better results for the detection of malaria disease. Statistical measurements such as precision, recall, f1-score, roc curve, etc. were calculated to validate the results. It can be concluded that this research work presents the state of art results when compared to the existing research. The work can be extended to explore the other deep learning techniques with different preprocessing techniques for image processing for better results.

## Figures and Tables

**Figure 1 diagnostics-13-00534-f001:**
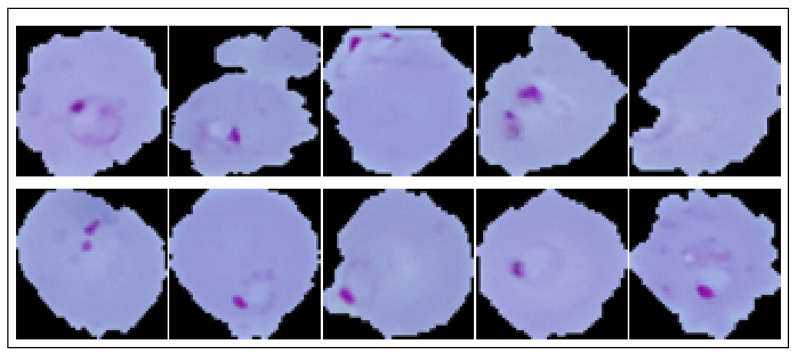
Infected cell images.

**Figure 2 diagnostics-13-00534-f002:**
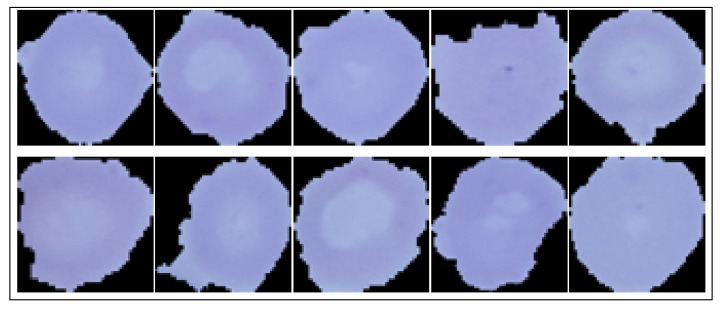
Uninfected cell images.

**Figure 3 diagnostics-13-00534-f003:**
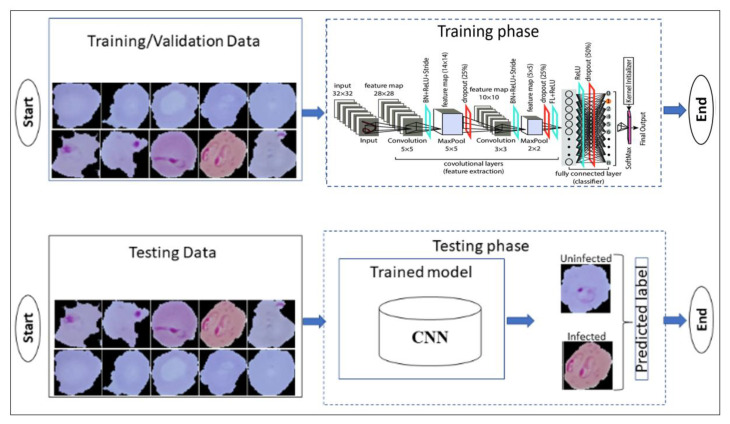
Block diagram of CNN model.

**Figure 4 diagnostics-13-00534-f004:**
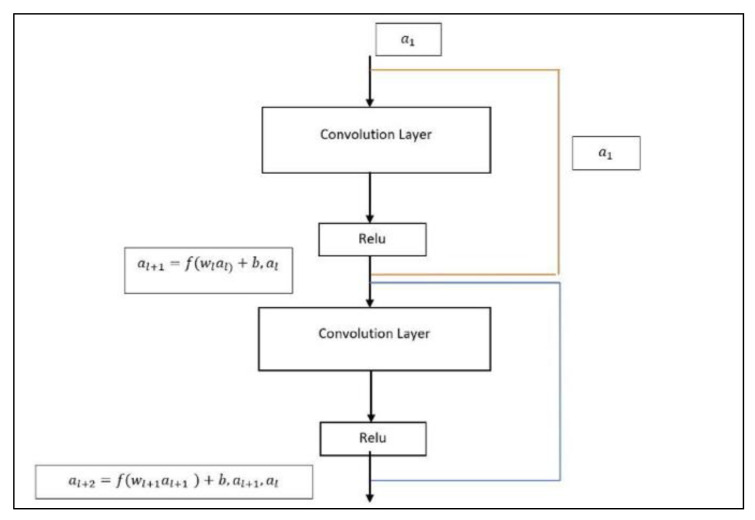
ReLU Algorithm.

**Figure 5 diagnostics-13-00534-f005:**
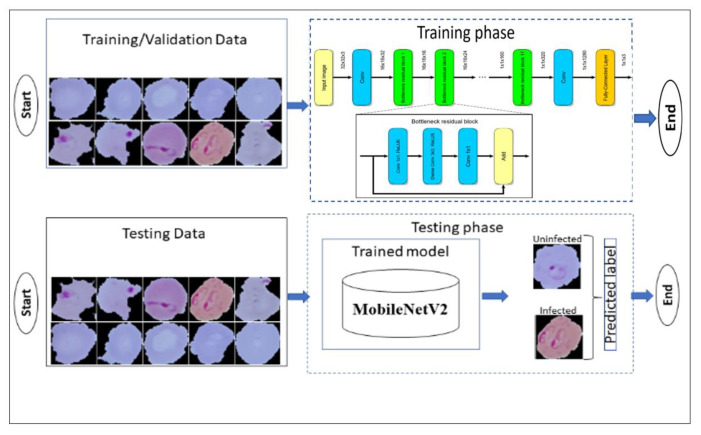
Block diagram of MobileNetV2 model.

**Figure 6 diagnostics-13-00534-f006:**
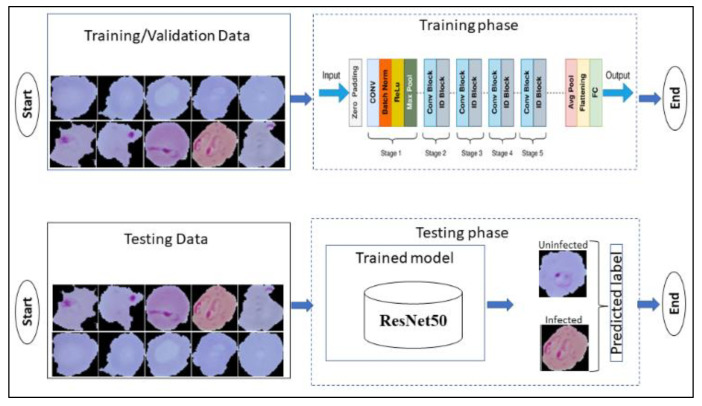
Block diagram of ResNet50 model.

**Figure 7 diagnostics-13-00534-f007:**
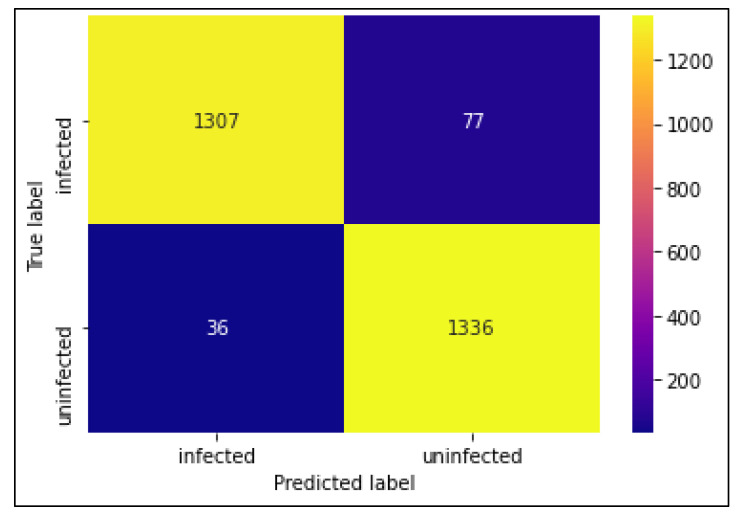
Confusion Matrix of CNN model.

**Figure 8 diagnostics-13-00534-f008:**
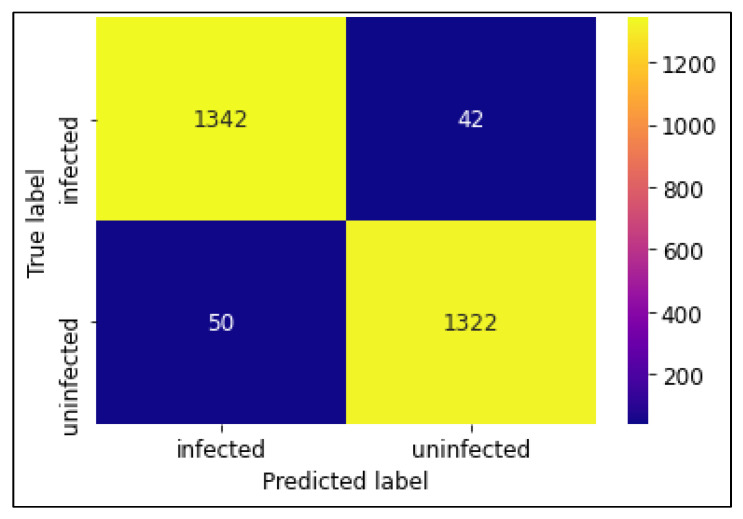
Confusion Matrix of MobileNetV2 model.

**Figure 9 diagnostics-13-00534-f009:**
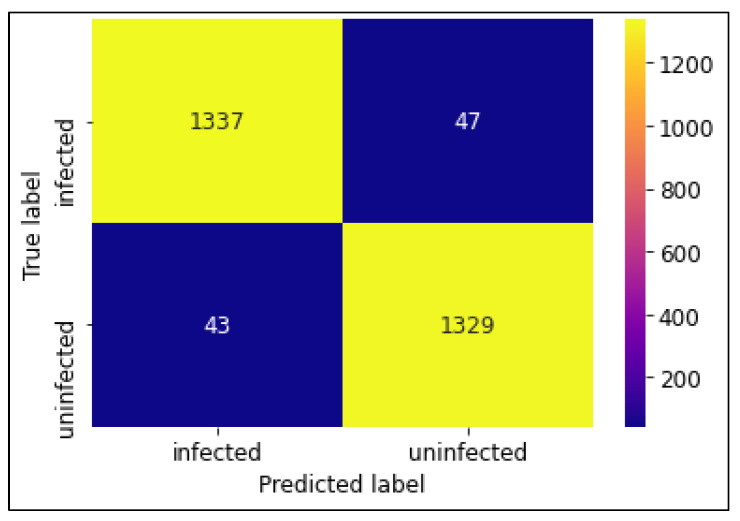
Confusion Matrix of ResNet50 model.

**Figure 10 diagnostics-13-00534-f010:**
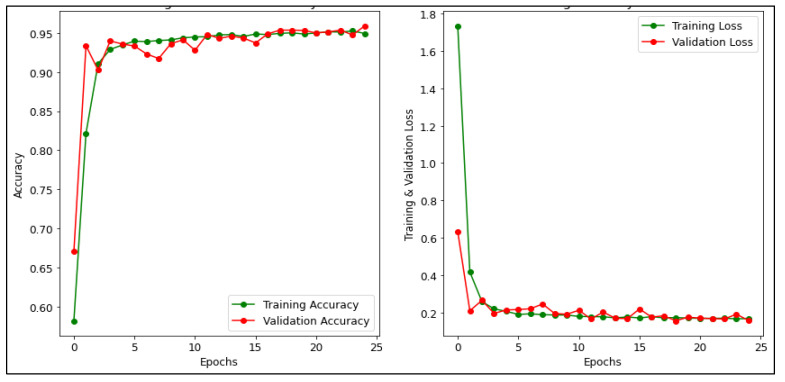
Accuracy and Loss of CNN model.

**Figure 11 diagnostics-13-00534-f011:**
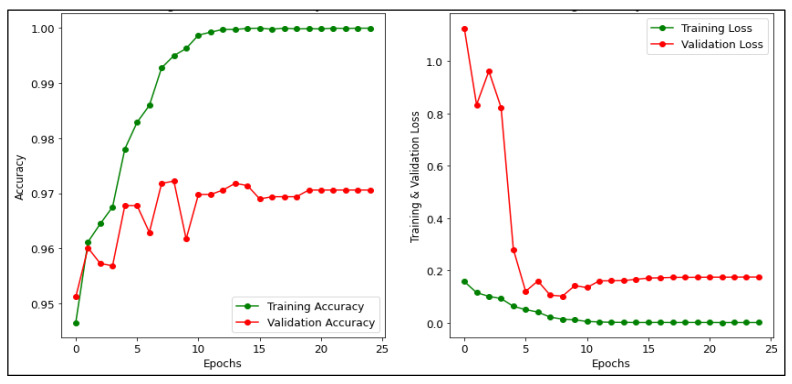
Accuracy and Loss of MobileNetV2 model.

**Figure 12 diagnostics-13-00534-f012:**
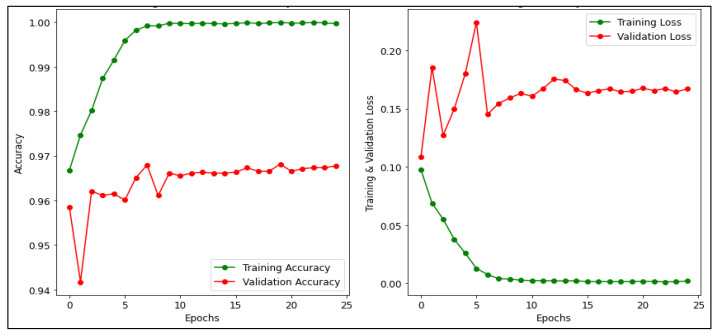
Accuracy and Loss of ResNet50 model.

**Figure 13 diagnostics-13-00534-f013:**
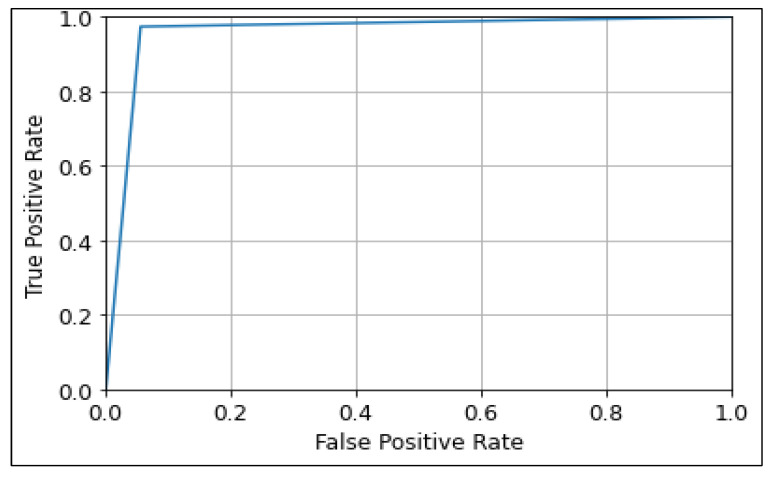
ROC Curve of CNN model.

**Figure 14 diagnostics-13-00534-f014:**
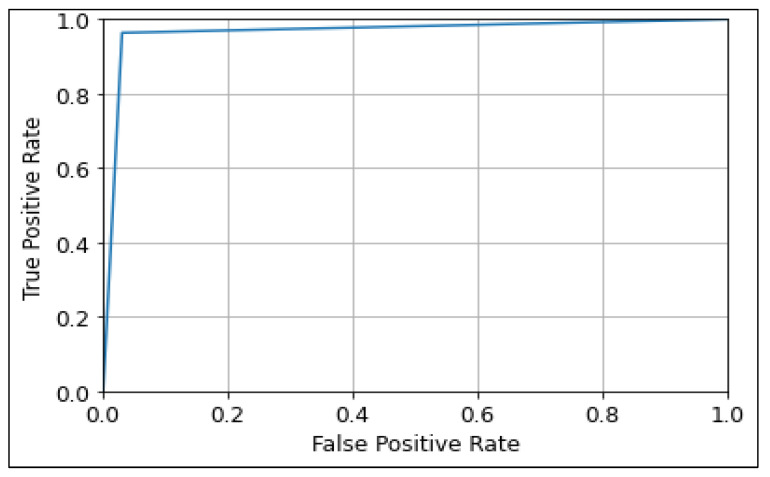
ROC Curve of MobileNetV2 model.

**Figure 15 diagnostics-13-00534-f015:**
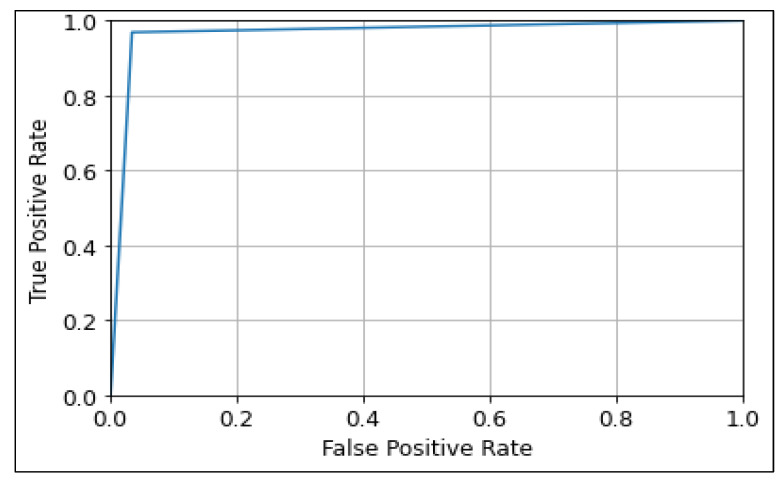
ROC Curve of ResNet50 model.

**Figure 16 diagnostics-13-00534-f016:**
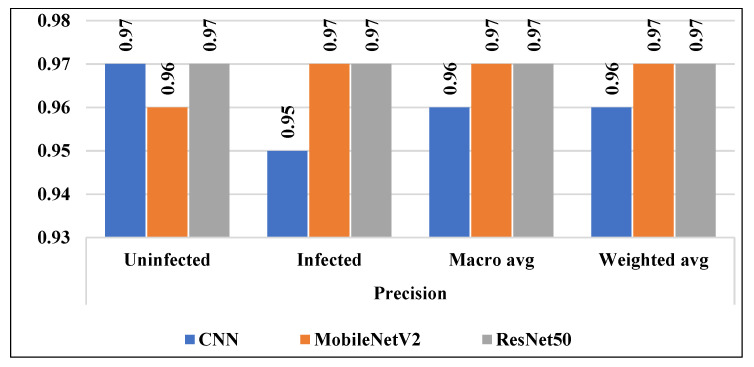
Precision of all considered models.

**Figure 17 diagnostics-13-00534-f017:**
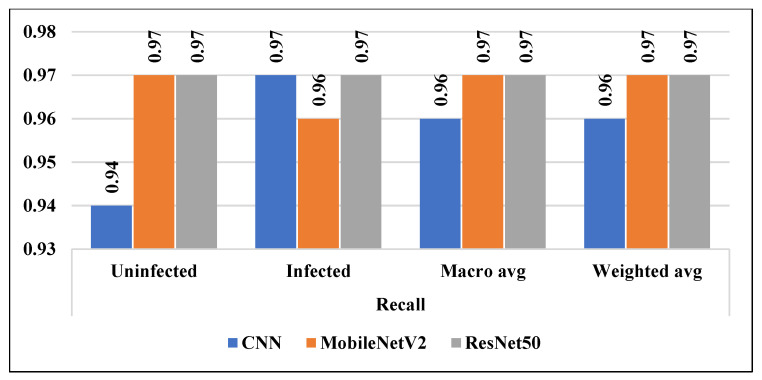
Recall of all considered models.

**Figure 18 diagnostics-13-00534-f018:**
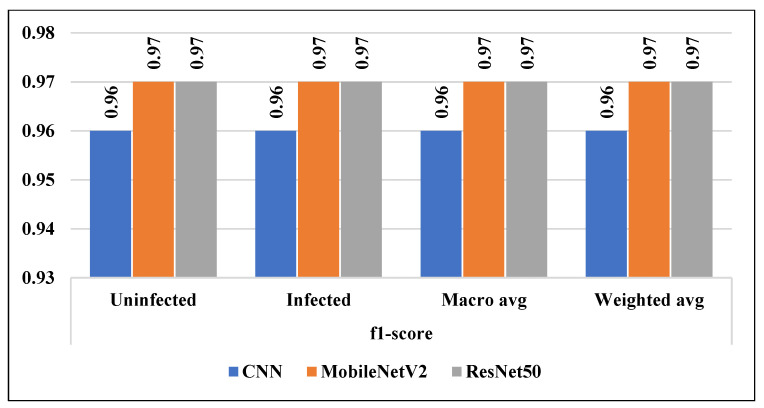
f1-score of all considered models.

**Table 1 diagnostics-13-00534-t001:** Convolutional neural Network model summary.

Layer (Type)	Output Shape	Param #
conv2d (Conv2D)	(None, 66, 66, 18)	504
max_pooling2d (MaxPooling2D)	(None, 33, 33, 18)	0
dropout (Dropout)	(None, 33, 33, 18)	0
conv2d_1 (Conv2D)	(None, 31, 31, 72)	11,736
max_pooling2d_1 (MaxPooling2D)	(None, 15, 15, 72)	0
dropout_1 (Dropout)	(None, 15, 15, 72)	0
conv2d_2 (Conv2D)	(None, 13, 13, 72)	46,728
max_pooling2d_2 (MaxPooling2D)	(None, 6, 6, 72)	0
dropout_2 (Dropout)	(None, 6, 6, 72)	0
conv2d_3 (Conv2D)	(None, 4, 4, 72)	46,728
max_pooling2d_3 (MaxPooling2D)	(None, 2, 2, 72)	0
dropout_3 (Dropout)	(None, 2, 2, 72)	0
flatten (Flatten)	(None, 288)	0
dense (Dense)	(None, 72)	20,808
dropout_4 (Dropout)	(None, 72)	0
dense_1 (Dense)	(None, 72)	146

## Data Availability

This dataset was taken from the National Institutes of Health (NIH) website (https://www.nih.gov/, accessed on 11 August 2022).

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
