# Peer review of "Performance Analysis of Deep Learning Algorithms in Diagnosis of Malaria Disease"

_diagnostics, 2023, doi:10.3390/diagnostics13030534_

Round 1

Reviewer 1 Report

I congratulate the authors for this article which adds knowledge to the field of artificial intelligence applied to the diagnosis of malaria on microscopic slides. The educational tendency of the article is certainly useful for a beginner audience.

Unfortunately ...

The introduction to the article could dwell less on the history of this field of research. You can keep these details for scientific reports other than an article (thesis document for example). The document looks like a technical report. The technical names of the models and calculation techniques are cited more than the methods themselves.

The work would be much more relevant if several key issues were addressed: (1) Evaluation of algorithms on new datasets directly from clinical practice; (2) Evaluation of the performances on unbalanced datasets. This is a real issue in a real-world context, since the data used from the NIH have a prevalence of positive slides, far above reality. (3) The notion of cross-validation during model learning, which is not clearly addressed. Moreover, some results are surprising like the ROC curves with only one cut-off point... Moreover, these curves should be put on the same graph to be easily compared. Perhaps if you have to talk about the state of the art of DL methods in AI-assisted diagnosis of malaria on lab slides, one of the papers below should be cited, especially since you seem to have drawn inspiration from one of them ...

Li S, Du Z, Meng X, Zhang Y. Multi-stage malaria parasite recognition by deep learning. Gigascience. 2021 Jun 17;10(6):giab040. doi: 10.1093/gigascience/giab040. PMID: 34137821; PMCID: PMC8210472.

1.       Maturana CR, de Oliveira AD, Nadal S, Bilalli B, Serrat FZ, Soley ME, Igual ES, Bosch M, Lluch AV, Abelló A, López-Codina D, Suñé TP, Clols ES, Joseph-Munné J. Advances and challenges in automated malaria diagnosis using digital microscopy imaging with artificial intelligence tools: A review. Front Microbiol. 2022 Nov 15;13:1006659. doi: 10.3389/fmicb.2022.1006659. PMID: 36458185; PMCID: PMC9705958.

2.       Uzun Ozsahin D, Mustapha MT, Bartholomew Duwa B, Ozsahin I.  Evaluating the Performance of Deep Learning Frameworks for Malaria Parasite Detection Using Microscopic Images of Peripheral Blood Smears. Diagnostics (Basel). 2022 Nov 5;12(11):2702. doi: 10.3390/diagnostics12112702. PMID: 36359544; PMCID: PMC9689376.

3.       Aimon Rahman, Hasib Zunair, Tamanna Rahman Reme, M. Sohel Rahman, M.R.C. Mahdy, A comparative analysis of deep learning architectures on high variation malaria parasite classification dataset

The scientific value of the article and its originality seem questionable.

Reviewer 2 Report

The authors compared three models of deep learning applied to the detection of malaria parasites: CNN, MobileNetV2, and ResNet50. The authors show that, among these three models, ResNet50 had the highest accuracy and least number of errors. The work is interesting and is an important contribution to the innovative methods of malaria diagnosis.

The manuscript needs more care and attention in terms of organization of ideas and many minor points to facilitate readers’ comprehension of the text, including English. The reference numbers, as well as Figure numbers, seem to be mixed up. Please re-check them. References need to be re-checked, and some need to be completed.

MAJOR COMMENTS:

Page 8, experimental results, “CNN’s are effective methods for identifying picture classes…” to the end of the paragraph “…The MobileNet concept aims to maximize computational power and memory use”: This part of the paragraph seems to be more appropriate in the Introduction because it does not refer to the authors’ results. Furthermore, exactly the same paragraph is in the Discussion (page 14).

MINOR COMMENTS:

Page 1, Introduction: I suggest - Malaria is a parasitic disease transmitted by Plasmodium (in italic) spp. Malaria due to Plasmodium falciparum is life-threatening…

Page 1, Introduction, “report by the World Health Organization (WHO)”: Please cite the WHO reference and update the data (instead of 2019, the latest available report provides an updated 2021 data): World Malaria Report 2022. (https://www.who.int/teams/global-malaria-programme/reports/world-malaria-report-2022)

Page 2, Introduction, “conventional methods of diagnosis…”: like microscopy, clinical diagnosis, and PCR. Why is microscopy not dependent on lab technician’s capacity?

Page 2, Introduction, “…as they are effective diagnostic methods for malaria”: The reason is not only because these methods are effective, but because microscopy and/or RDT are available in the field, even in remote villages.

Page 2, Introduction, “microscopic systems…considered an effective diagnostic method as there are no shortcomings as in RDT”: The meaning of this statement is not clear. The interpretation of microscopy is ‘subjective,’ and its effectiveness is dependent on the expertise of the microscopist. Please clarify.

Page 2, Introduction, “…classifies the infected cells [14] & [19]”: If the journal requires reference citation numbering in the order of citation, these references should be 4 and 5. Please re-check reference numbering.

Page 2, Introduction, “in 2018, Md Zahangir Alom surveyed deep learning approaches”: In REF 8, the author is referred to as “M.Z. Alom.” Place a period after “approaches” and start a new sentence for “In that survey…”

Page 2, Introduction, “Bayesian, linear regression ith Relu…”: Please check the reference numbering. REF 13 to 17 and 19-20 do not seem to have been cited yet.

Page 2, Introduction, “in 2019, malaria disease is predicted using the CNN model”: was predicted

Page 2, Introduction, “In 2019, malaria disease is predicted using the CNN model… Through mobile applications, malaria disease was predicted using CNN.”: I am somewhat confused with these two sentences. Do they refer to two different studies performed in 2019? Please clarify.

Page 2, Introduction, “Yuhang Dong 2017…”: delete “2017”

Page 2, Introduction, “…machine learning and computer vision”: Period here, then a new sentence “These tools provide accurate and efficient computation (not “computationally”)”

Page 2, Introduction, “employing hyperparameters…”: the use of… (or employment of…); validation of different performance metrics to identify the reliability and feasibility of the proposed framework

Page 2, Introduction, “The remaining portions of the paper…” to “concludes its findings”: This paragraph can be deleted.

Page 2, Related works, “Diagnosis of malaria is done through…”: diagnosis of malaria can be done through…

Page 3, Related works, “malaria, spread by Plasmodium parasites is a life-threatening disease, by the analysis of microscopic blood smear images, malaria is detected by a trained microscopist”: This sentence is not correctly constructed and is a repetition. It can be deleted.

Page 3, Related works, “Quinn et al [11] and Rosado et al [17]…”

Page 3, Related works, “Rosado et al. have evaluated/assessed/studied the recognition of malaria parasites”

Page 3, Related works, “98.2% and 72.1% for WBC, respectively”

Page 3, Related works, “the model’s accuracy and performance were optimized

Page 3, Related works, “with the help of the deep learning model used for detection accuracy of 99.23% experienced”: Please write a complete sentence.

Page 3, Related works, “Liang et al. achieved…[6]”: REF 6 is Kuo CCJ. Please check the reference number.

Page 3, Related works, “BDNs give a more flexible displaying system…” Please re-write the sentence in a correct syntax, especially the second half of this sentence.

Page 3, Related works, “Alex Krizhevsky and his team…”: Please cite a reference here. Is it [27-30]?

Page 3, Related works, “this ensemble method is a combination… variance and overfitting of the model were reduced”

Page 3, Related works, “See identified that uRDDT Alere Malaria…”:  Please check the meaning of this sentence. It also seems to be out of place.

Page 3, Related works, “But he faced difficulties…” “The accuracy was fine”

Page 3, Related works, “microscopic examination of Giemsa-stained blood films is very efficient in Plasmodium (capital letter “P” and in italics) detection” “discussed the advantages and disadvantages of other diagnostic methods…

Page 4, Related works, “In this study the fundamentals…Africa [40]”: Please write a complete sentence with a subject and a verb.

Page 4, Related works, “Plasmodium falciparum” and “P. falciparum” in italics

Page 4, Related works, “These results showed that some children (not “kids”)…”

Page 4, Related works, at the end of the sub-section “Related works”: I think that the authors should add a sentence or two to conclude this section on the overview of relevant works.

Page 4, Materials and methods, NIH website: Please provide the website.

Page 4, Materials and methods, “the infected and uninfected cell images are shown in figures 2 and 3”: Should be Figures 1 and 2.

Page 5, Materials and methods, “Figure 1. Uninfected cell images”: should be Figure 2.

Page 6, Materials and methods, “The image is convolved with a kernel or filter while the transformation process.” Please complete the sentence. There is something missing here.

Page 6, Materials and methods, ReLU algorithm – this should be Figure 4. 

Page 6, Materials and methods, “where the convolution layer is the primary component of CNN”: This is not a complete sentence.

Page 7, Materials and methods, “the output at this stage is given by the equations”: The output at this stage is given by the equations (1) and (2), and this process is repeated…?

Page 7, Materials and methods, “RESNET is having…”: RESNET has…; ResNet50 as shown in Figure 6 was used…

Page 8, Materials and methods, block diagram of ResNet50 model – should be Fig. 6

Page 8, experimental results, “malaria is a serious disease that continues to cause concern around the world”: A similar statement was made in the introduction. It was also repeated at the beginning of the Discussion (bottom of page 13). This sentence can be deleted.

Page 8, experimental results, “The classification performances of the algorithms are evaluated…”: The Results section should be in general written in the past tense: The classification performances of the algorithms were evaluated. The same remark for the entire section on experimental results.

Fig 16-18: These figures may require a legend that explains what “macro avg” and “weighted avg” mean.

Page 14, Discussion, three paragraphs, from “Fig. 10 presents the accuracy and loss for training and validation of the CNN model” to “in terms of f1-score, MobileNetV2 and ResNet50 attained a rate of 0.97%”: These three paragraphs are comments on the Figures presented in the Results section. They may be more appropriately placed in the Results section.

Page 14, Discussion, “Also, an AUC value closer to Also, an AUC value closer to 1 denotes…: Delete the first “Also, an AUC value closer to” which is repeated.

Page 15, Conclusions, “…that environmental elements are crucial in facilitating. Malaria’s presence and transmission”: Delete the period after “facilitating” and “malaria” (with small letter “m”)

Page 15, Conclusions, “Among all the models, ResNet50 outperformed and provide better results…”: provided

REF 1 is incomplete. The title of the supplement does not have to be cited. Here is how Pubmed records this publication: Wongsrichanalai C, Barcus MJ, Muth S, Sutamihardja A, Wernsdorfer WH. A review of malaria diagnostic tools: microscopy and rapid diagnostic test (RDT). Am J Trop Med Hyg. 2007;77(6 Suppl):119-127.

REF 4: Please check the authors. If there are many authors, “et al.” can be used. This comment also refers to other references (for example, REF 14). This reference (REF 4) seems to be incomplete. Please provide more information (web link, for example).

REF 8: This is a pre-print. If this paper was published since 2018, please provide the updated reference. Same with REF 24.

REF 9: Please check if this reference is complete.

REF 35 seems to be incomplete.

REF 43: Applied Artificial Intelligence, 36:1

REF 44: The correct citation of this REF is as follows: Marques G, Ferreras A, de la Torre-Diez I. An ensemble-based approach for automated medical diagnosis of malaria using EfficientNet. Multimed Tools Appl. 2022;81(19):28061-28078.

Round 2

Reviewer 1 Report

Although the originality and the part of innovation in this article remains to be proven, it can however be published if one considers its didactic and pedagogical value.

I don't feel qualified to judge about the English language and style

Best regards